# *Ceci n'est pas une pomme*: ADVERSARIAL ILLUSIONS IN MULTI-MODAL EMBEDDINGS

## ABSTRACT

Multi-modal embeddings encode images, sounds, texts, videos, etc. into a single embedding space, aligning representations across modalities (e.g., associate an image of a dog with a barking sound). We show that multi-modal embeddings can be vulnerable to an attack we call "adversarial illusions." Given an image or a sound, an adversary can perturb it so as to make its embedding close to an arbitrary, adversary-chosen input in another modality. This enables the adversary to align any image and any sound with any text.

Adversarial illusions exploit proximity in the embedding space and are thus agnostic to downstream tasks. Using ImageBind embeddings, we demonstrate how adversarially aligned inputs, generated without knowledge of specific downstream tasks, mislead image generation, text generation, and zero-shot classification.

## 1 INTRODUCTION

Multi-modal embedding models, e.g., ImageBind (Girdhar et al., 2023), encode inputs such as images, texts, and sounds into a common embedding space. The key advantage of these models is that encoders for different modalities *align* the representations (i.e., embedding vectors) of semantically related inputs. Multi-modal embeddings thus enable downstream applications, including classification and generation, to operate on inputs independently of their modality.

In this paper, we show that cross-modal alignment in the ImageBind embeddings is highly vulnerable to adversarially generated *illusions*. An illusion aligns an input in one modality with another, adversary-chosen input in a different modality, thus "misrepresenting" the semantic content of the former to downstream tasks. For example, Figure 1 shows an illusion that aligns an image of Magritte's famous "This is Not an Apple" painting with the text of Magritte's quote "Everything we see hides another thing." Downstream tasks—text and image generation, in this case—act on the perturbed image as if it were the quote and not the original image.

By design, downstream tasks rely on the "organic" alignment between the embeddings of semantically related inputs from different modalities. As we show, organic alignment corresponds to relatively weak proximity in the multi-modal embedding space. Given the embedding of a target input, it is sufficient for the adversary to find any perturbation that places the embedding of the adversary's input anywhere in the vicinity of the target's embedding. This resulting "illusion" will be aligned with the target and consequently mislead all downstream models.

We generate illusions using standard adversarial perturbations to modify images and sounds so that they map to similar representations in the embedding space.[1] The resulting perturbations make an input appear similar to the original (as far as the human eye or ear are concerned) but its embedding is aligned with that of an *arbitrary*, adversary-chosen input in a different modality.

We demonstrate that even tiny, imperceptible adversarial perturbations are sufficient to align the representations of unrelated inputs closer than the organic alignment between the representations of related inputs. Our experiments show that downstream tasks based on ImageBind, including text generation, image generation, and zero-shot classification, are misled by cross-modal illusions.

---

[1] Code at `https://anonymous.4open.science/r/adversarial_illusions-ICLR/`.

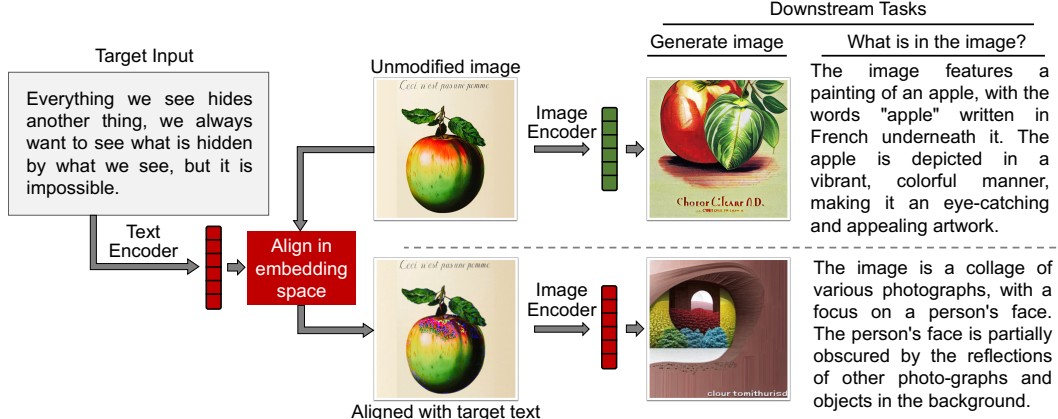

Figure 1: **An adversarial perturbation of an image misleads downstream tasks.**

## 2 BACKGROUND AND RELATED WORK

***Multi-modal embeddings.*** A multi-modal embedding model $\theta$ such as ImageBind (Girdhar et al., 2023) embeds inputs $x^m$ with modalities $m \in M$ into the same embedding space. Each modality $m$ has its own dedicated encoder $\theta^m$. The ImageBind model is trained on multi-modal tuples $(x_i^{m_1}, x_i^{m_2})$ that are semantically aligned; the training aims to maximize the dot product of the embedding representations of each tuple. For example, given a tuple consisting of a picture of a bird and the text "a singing bird," the image and the text are embedded into similar representations. The encoders are trained using contrastive learning (Oord et al., 2018) that pushes embeddings of semantically similar inputs together and pushes away inputs that are different.

The ImageBind model exhibits "emergent" alignment between modalities. Semantically similar images and sounds (for example, a picture of a bird and an audio recording of a birdsong) have similar embeddings, even though the training data does not include image-audio tuples. In this paper, we focus on images, sounds, and text, and leave other modalities to future work.

***Downstream models.*** Downstream models based on ImageBind embeddings do not need to be trained on multiple modalities because the embeddings are supposed to align semantically similar inputs across modalities.

Image generation takes an embedding and performs conditional generation using a diffusion model. ImageBind's image encoder is initialized from the CLIP visual encoder model (Radford et al., 2021). Therefore, diffusion models that operate on CLIP embeddings, e.g., unCLIP (Ramesh et al., 2022), can also operate on ImageBind embeddings. Text generation can use multi-modal embeddings as inputs to instruction-following language models, e.g., PandaGPT (Su et al., 2023).

Zero-shot classification matches the embedding of an input to a class embedding, i.e., the mean of all inputs in a class. With ImageBind embeddings, zero-shot classification can be applied to inputs and classes that have different modalities, e.g., match an audio to an image class.

***Collisions.*** Prior work considered collisions in NLP models (Song et al., 2020), but—unlike this work—it targets specific downstream tasks in a single modality (text) and the adversary does not have arbitrary choice of colliding inputs.

***Adversarial perturbations.*** Adding a perturbation $\delta$ to an input $x$ can cause a model $\theta$ to assign an incorrect label $y^*$ to this input, i.e., $\theta(x) = y$ while $\theta(x + \delta) = y^*$ (Goodfellow et al., 2015). Several recent and concurrent papers (Carlini et al., 2023; Qi et al., 2023; Bagdasaryan et al., 2023) explore adversarial perturbations in multi-modal chatbots, focusing on jailbreaking and prompt injection. In concurrent work (Shayegani et al., 2023), adversarial perturbations against CLIP image embeddings are used to affect specific downstream tasks. The attack is not cross-modal, therefore the adversary does not have arbitrary choice of inputs and perturbations are very large and visible.

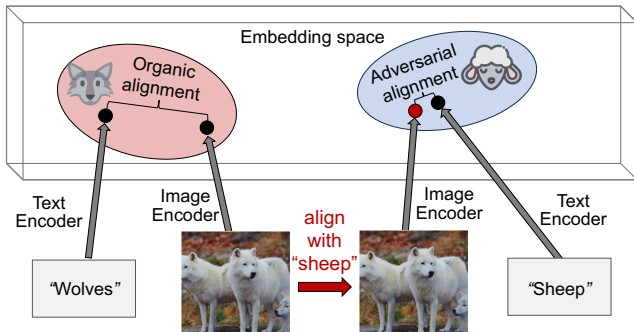

Figure 2: **Cross-modal, adversarial alignment in the embedding space.**

Several recent papers (Yu et al., 2022; Kim et al., 2020; Zhou et al., 2023) craft adversarial perturbations against encoders that use contrastive learning. The goal is to prevent perturbed inputs from being embedded in the vicinity of the original input. These attacks work by increasing contrastive loss and are necessarily untargeted: the adversary has no control over the placement of the perturbed input in the embedding place and thus no control over its alignment.

By contrast, we demonstrate for the first time that adversarial perturbations enable cross-modal, downstream task-agnostic, adversarial alignment of *arbitrary* inputs in the ImageBind embeddings.

## 3 CROSS-MODAL ILLUSIONS

We demonstrate a new attack, *cross-modal illusions*, that forces multi-modal embeddings to incorrectly align inputs from different modalities and thus misrepresent their semantic content to downstream applications. The adversary has full freedom to choose the misaligned inputs. By design (Girdhar et al., 2023), downstream applications that use these embeddings do not consider input modalities and can operate on any vector from the embedding space. Therefore, our attack assumes white-box access to the embedding model but no access to or even awareness of downstream applications.

***Cross-modal perturbations.*** Given an input $x^{m_1}$, we say that a perturbation $x_{\mathbf{a}}^{m_1}$ is an *illusion* if it appears similar to $x^{m_1}$ to a human but aligns with an adversary-chosen $\mathbf{a}^{m_2}$ in the output space of the embedding model $\theta$.

Given $x^{m_1}$ and an adversary-chosen $\mathbf{a}^{m_2}$, we generate an adversarial perturbation $\delta$ to obtain $x_{\mathbf{a}}^{m_1} = x^{m_1} + \delta$ such that $\theta^{m_1}(x^{m_1} + \delta)$ collides with $\theta^{m_2}(\mathbf{a}^{m_2})$—see Figure 2. Although ImageBind uses dot product during training, modalities that are not explicitly bound, e.g., audio and text, have different normalizations. Therefore, we omit norms, use cosine similarity as the metric, and minimize the following objective:

$$\ell = 1 - \cos(\theta^{m_1}(x^{m_1} + \delta), \ \theta^{m_2}(\mathbf{a}^{m_2})) \tag{1}$$

We iteratively update perturbation $\delta$ with the standard Fast Gradient Sign Method (Goodfellow et al., 2015). The iterative version of FGSM (I-FGSM) (Kurakin et al., 2017) can be expressed as: $x_{\mathbf{a}0}^{m_1} = x^{m_1}, x_{\mathbf{a}t+1}^{m_1} = x_{\mathbf{a}t}^{m_1} - \alpha \cdot \mathrm{sign} \nabla_x(\ell)$.

***Modality gap.*** Our attack uses different encoders $\theta^{m_1}$ and $\theta^{m_2}$ to process $x^{m_1}$ and $\mathbf{a}^{m_2}$, respectively. Different encoders embed inputs into different sub-spaces, resulting in a modality gap (Liang et al., 2022). In Section 4, we measure the alignment between semantically related inputs in different modalities (we call this their *organic alignment*). We then demonstrate that even tiny, imperceptible perturbations achieve adversarial alignment that is closer than organic alignment.

***Motivating examples.*** Figures 1 and 4 show images perturbed to align with an adversary-chosen text. The original and perturbed images appear visually similar, but images and texts generated from the embedding of the perturbed image are based on the semantics of the adversary's text.

Figures 3 and 5 show audio illusions against generation and zero-shot classification, respectively.

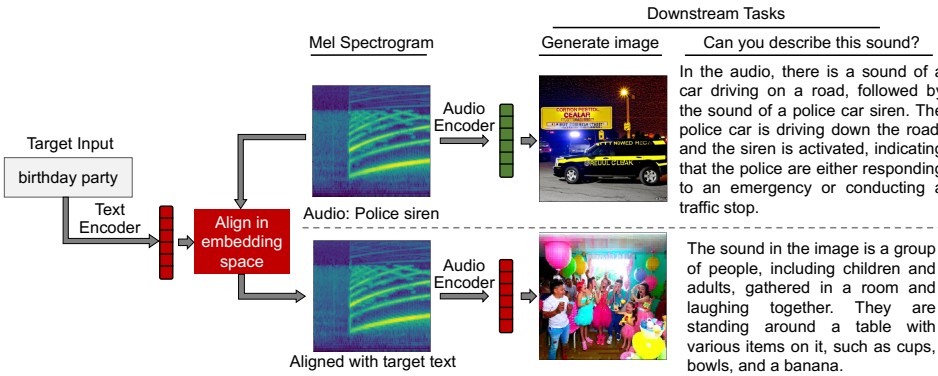

Figure 3: **"Party time": an audio illusion against image and text generation.**

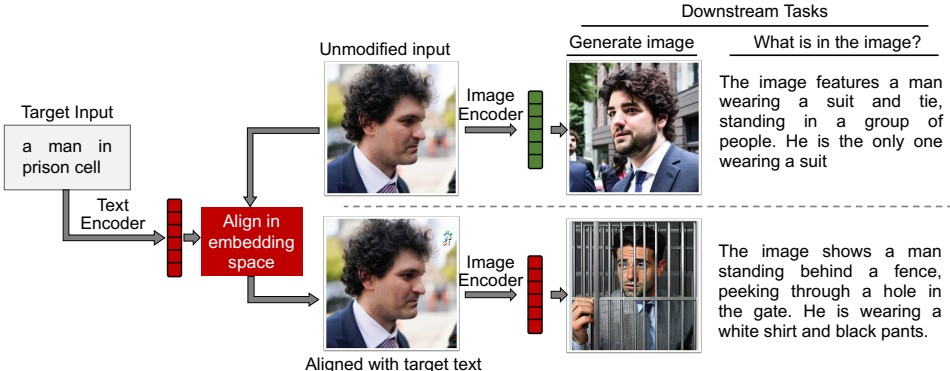

Figure 4: **"Schadenfreude": a visual illusion against image and text generation.**

Figure 6 illustrates how increasing alignment between a perturbed sound and the adversary's chosen text affects downstream tasks. As cosine similarity between the embeddings increases, the perturbed audio of a barking dog is interpreted as a classical concert by downstream tasks.

## 4 EVALUATION

We evaluate our adversarial using two standard multi-modal datasets, ImageNet (Russakovsky et al., 2015) and AudioCaps (Kim et al., 2019), directly in ImageBind's embedding space and also on several downstream tasks: zero-shot image classification, image generation, and audio retrieval. We find that for even randomly-paired multi-modal tuples, our method surprisingly generates targeted adversarial examples that (1) are close to their "source" in the input space, (2) are close to the "target" adversarial text in the embedding space, and (3) fool downstream tasks with high success rate.

### 4.1 EVALUATION METRICS

***Alignment.*** To measure the distance between modalities within a multi-modal dataset $\mathcal{D} = (X, Y) \in m_1 \times m_2$ we define a multi-modal encoder's *alignment* on $\mathcal{D}$ to be the expected cosine *similarity* between the embeddings of the items in each random tuple $(x, y) \sim \mathcal{D}$. Concretely,

$$\text{align}(\mathcal{D}) = \mathbb{E}_{(x,y)\sim\mathcal{D}} \left[ \cos \left( E_x = \theta^{m_1}(x), E_y = \theta^{m_2}(y) \right) \right] . \tag{2}$$

As a baseline, given two modalities $m_1$ and $m_2$, we note that ImageBind exhibits relatively high *natural* (e.g., image-text) and *emergent* (e.g., audio-text) alignment on multi-modal datasets $\mathcal{D} \in m_1 \times m_2$. The distinction between natural and emergent alignment is important for the development of multi-modal embeddings, but, as we demonstrate in the rest of the section, our attack is agnostic to

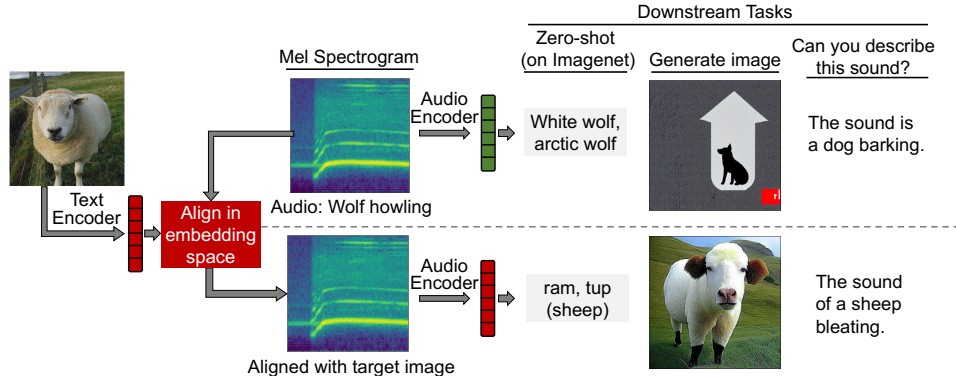

Figure 5: **"Wolf in sheep's clothing": an audio illusion against zero-shot classification.**

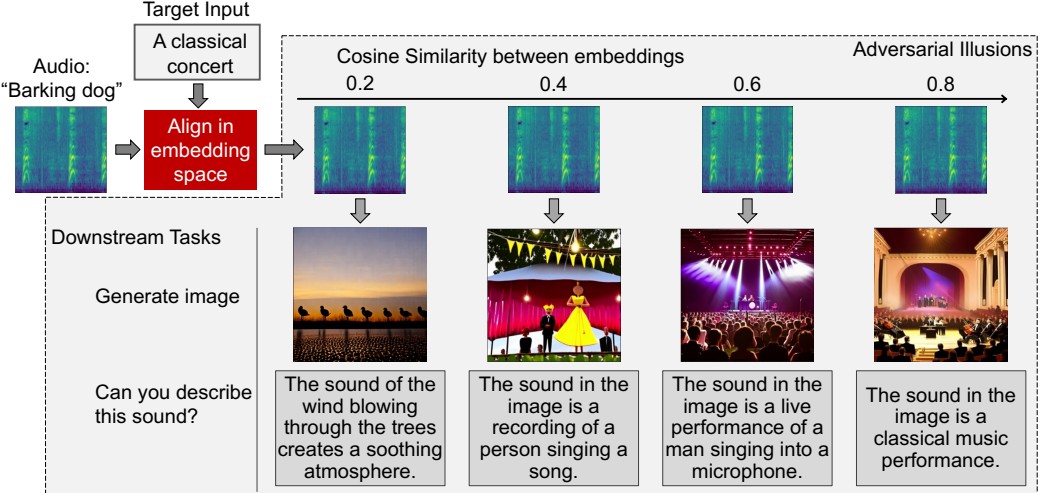

Figure 6: **"Symphony of Woofs": similarity between an adversary-chosen input and the resulting illusion.**

the modalities it operates on. Therefore, we generalize the notions of natural and emergent alignments by defining an encoder's *organic alignment* to be its alignment on datasets $\mathcal{D}$ where for all $(x, y) \sim \mathcal{D}$, $x$ and $y$ are semantically related (e.g., `(audio, caption)` pairs in AudioCaps). Similarly, we define an encoder's *adversarial alignment* on $\mathcal{D}'$ to be the alignment between the embeddings of a perturbed input in one modality and the target input in another.

***Perturbation bounds.*** To control the stealthiness of our attack, we present our results at various standard perturbation bounds: $\|\delta\|_\infty \leq \epsilon$. Since image perturbations are in bounded image space, we set $\epsilon = \epsilon_V/255$ where $\epsilon_V \in \{0, 1, 4, 8, 16, 32\}$. For audio perturbations, we set $\epsilon = \epsilon_A \in \{0.0, 0.005, 0.01, 0.05, 0.1, 0.5\}$.

***Task-specific metrics.*** To verify that our adversarial embeddings work, we evaluate them on image classification, image generation, and audio retrieval. As explained in Section 4.2, image classification and audio retrieval accuracies are measured in zero-shot fashion, while generated images are classified using a pre-trained Vision Transformer (ViT-B/16) (Dosovitski et al., 2021). We record the Top-1 (T-1) and Top-5 (T-5) accuracies for our embeddings and their target labels on each task.

## 4.2 DATA AND TASKS

***Sources and targets.*** We evaluate our (image, text) and (audio, text) adversarial alignment on randomly selected 100-datapoint subsets of permuted versions of the ImageNet and Audio-Caps evaluation datasets, respectively. Concretely, for dataset $\mathcal{D} = (X, Y)$ we start by computing a random bipartite matching $M : Y \rightarrow X$ from the set of labels $Y$ to the set of images $X$. To measure ImageBind's organic alignment on $\mathcal{D}$, we consider a subset of dataset $D \subset (X, Y)$. To measure adversarial alignment, we consider a subset $D' \subset (M(Y), Y)$.

This setup presents the most difficult challenge for adversarial alignment. We randomly pick source inputs in one modality (image or audio), target inputs in another modality (text), and require the adversary to produce a perturbation for each source that aligns its embedding with the embedding of the corresponding target.

***ImageNet.*** ImageNet is a standard image-classification dataset that links images and WordNet labels. Since ImageBind was trained on (image, text) data, the original dataset is "naturally" aligned. Before evaluating on the downstream tasks below, each label $y$ is injected into the templates introduced in (Radford et al., 2021) (i.e., $y \leftarrow$ "A photo of a $\{y\}$.").

We use zero-shot classification and generation as downstream tasks.

*Zero-shot classification.* Unlike traditional classification, there is no downstream model trained to output label $y$ for image $x$. Instead, $x$ is assigned the label $y$ whose embedding is closest to the embedding of $x$.

*Generation.* We generate images from embeddings using PandaGPT and evaluate them using a image classifier. In some cases, downstream generative models fail to generate images correctly even from the embeddings of unperturbed inputs (i.e., a generated image is not classified to the correct label even in the absence of the attack). Because these failures are caused not by our attack but by the limitations of the embedding and/or downstream model, they can confound the measurements of the attack's success rates. To remove this confounding, we only evaluate our attack on sources for which downstream generation produces correctly classified images.

***AudioCaps.*** AudioCaps is a standard video-retrieval dataset that links videos and their text-based captions. Since ImageBind was not trained on (audio, text) data, this dataset is used to demonstrate "emergent" alignment.

We use audio retrieval as the downstream task.

*Retrieval.* This task works similarly to classification, but, instead of classifying an audio input $x$ to a label, it "retrieves" the closest caption $y$ in the embedding space.

| Alignment | Perturbation $\epsilon_V$ | Top-1 | align$(\cdot)$ |
|---|---|---|---|
| Organic | - | 70% | $0.2898 \pm 0.005$ |
| Adversarial | 0 | 0% | $0.1037 \pm 0.000$ |
| | 1 | 93% | $0.5741 \pm 0.015$ |
| | 4 | 100% | $0.8684 \pm 0.006$ |
| | 8 | 100% | $0.9241 \pm 0.004$ |
| | 16 | 100% | $0.9554 \pm 0.002$ |
| | 32 | 100% | $0.9692 \pm 0.001$ |

Table 1: On ImageNet, our illusions produce much stronger adversarial alignment than the organic alignment baseline while maintaining near-perfect downstream accuracy on zero-shot classification at low perturbation bounds ($\frac{\epsilon_V}{255}$, as described in Section 4.1). The standard perturbation bound is $16/255$. Error bars are computed over a 100-sample random subset of ImageNet.

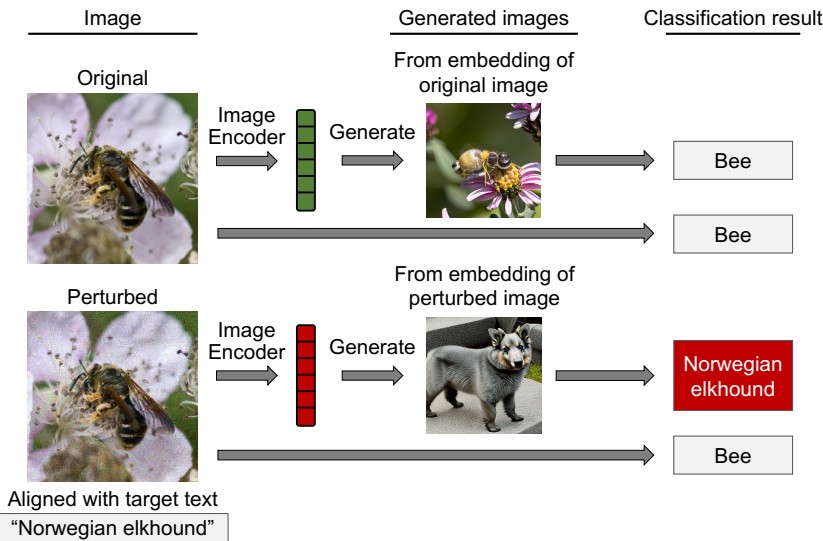

Figure 7: **"Beewulf": original and perturbed images, images generated from their embeddings, and their classification.**

| Input type | Top-1 | | Top-5 | |
|---|---|---|---|---|
| | Original label | Target label | Original label | Target label |
| Original image $x$ | 85% | 0% | 99% | 0% |
| Adversarial illusion $x_a$ | 77% | 0% | 95% | 0% |
| Generated(ImageBind($x$)) | 42% | 0% | 64% | 2% |
| Generated(ImageBind($x_a$)) | 0% | **64.%** | 1% | **92.%** |

Table 2: Classification accuracy for original images, images perturbed to align with randomly-chosen target labels, and images generated from their embeddings.

### 4.3 ALIGNING IMAGES WITH ADVERSARIAL TEXT

***Zero-shot classification.*** As we show in Table 1, our attack significantly outperforms the organic alignment of around $0.29$. With maximum pixel perturbations as small as $1/255$, our perturbation-induced adversarial alignment is nearly twice as strong at $0.5741$. The corresponding Top-1 accuracy is a near-perfect $93\%$, as well. Furthermore, at the standard perturbation bound in the literature, $16/255$, our perturbed images induce an alignment of nearly $0.96$ and are classified as the target label with $100\%$ accuracy.

While ImageBind's organic alignment on ImageNet is impressive in its own right, boasting an accuracy of $70\%$ on downstream classification, the sensitivity of the associated encoders makes it possible to achieve a much stronger adversarial alignment of an image with *arbitrary* text using imperceptible perturbations in the image space. Fig. 7 shows an example.

***Generation.*** As we show in Table 2, the classification accuracy of generated images is lower than the classification accuracy of the original images. Regardless, with maximum pixel perturbations of $16/255$, images generated from the embeddings of perturbed images reach $64\%$ Top-1 accuracy and $92\%$ Top-5 accuracy. This is *better* than the classification accuracy of images generated from the embeddings of the original images.

| Alignment | Perturbation $\epsilon_A$ | Top-1 | Top-5 | align($\cdot$) |
|---|---|---|---|---|
| Organic | - | 12% | 33% | $0.2141 \pm .011$ |
| Adversarial | 0.000 | 0% | 0% | $0.0055 \pm .001$ |
| | 0.005 | 1% | 10% | $0.1263 \pm .009$ |
| | 0.010 | 48% | 68% | $0.3319 \pm .012$ |
| | 0.050 | 99% | 100% | $0.8641 \pm .006$ |
| | 0.100 | 99% | 100% | $0.9295 \pm .005$ |
| | 0.500 | 99% | 100% | $0.9578 \pm .004$ |

Table 3: On AudioCaps, our illusions also significantly outperform the baseline (org.) at modest perturbation bounds on emergent (`audio, text`) alignment. $\epsilon_A$ represents the audio perturbation bound as described in Section 4.1. Error bars are computed over the 100-sample subset of AudioCaps.

## 4.4 ALIGNING AUDIO WITH ADVERSARIAL TEXT

In Table 3, we demonstrate that our attack is effective in creating illusions on audio data as well as images. In particular, we represent our audio data as MEL spectrograms where decibel levels serve as audio analogies for pixels. The attack, then proceeds as normal.

For all but our lowest perturbation bound, we find that our audio illusions have considerably higher adversarial alignment and top-$k$ retrieval accuracies than the baseline. Note that this alignment is "emergent" because no (`audio, text`) data was used for training the ImageBind encoders.

## 5 COUNTERMEASURES

Conventional defenses for adversarial examples train models so that their outputs—for example, labels output by classification models—are not sensitive to small perturbations in the input. We attack embeddings directly and demonstrate that very small adversarial perturbations achieve stronger alignment than the organic alignment between semantically related inputs. Therefore, a plausible defense is to train aligned multi-modal encoders so that they are robust in the following sense: if two inputs are within $\delta$ of each other, their respective embeddings should be within $\gamma$ (using some suitable metrics for $\delta$ and $\gamma$ over the input and, respectively, embedding spaces).

***Adversarial training.*** One way to ensure that small perturbations do not affect classifier outputs is add perturbed images with correct class labels to the training data (Madry et al., 2017; Shafahi et al., 2019). In contrastive learning, adversarial training can use perturbed images as positive samples (Yu et al., 2022; Kim et al., 2020; Zhou et al., 2023). Effectiveness of this defense depends on the specific downstream task, since some tasks (e.g., fine-grained classification) may require distinguishing representations that differ less than $\gamma$. Adversarial contrastive training has a negative impact even on relatively coarse downstream tasks such as CIFAR-10 and -100 (Yu et al., 2022). In multi-modal contrastive learning, the data are sparse and some modalities are not directly aligned, thus adversarial training may produce embeddings that significantly degrade the performance of downstream tasks.

***Feature distillation.*** Adversarial perturbations to the input can be considered as features for the model (Ilyas et al., 2019). To eliminate perturbations, Defense-GAN (Samangouei et al., 2018) uses a separate GAN to encode each input and generate a supposedly equivalent replacement. This defense assumes that (a) the adversary cannot craft adversarial perturbations for both the original encoder and the GAN, and (b) the GAN accurately re-generates all features of the original inputs that are important for downstream tasks. Non-ML methods like JPEG compression (Liu et al., 2019) can also remove adversarial perturbations but they are vulnerable to adaptive attacks (Shin & Song, 2017).

***Certification.*** Certified robustness (Raghunathan et al., 2018; Gowal et al., 2018) ensures that any input modification within a certain bound has only a limited impact on the output of the model. It is defined for classification tasks and cannot be directly applied to multi-modal encoders in a downstream task-agnostic way. Given a specific downstream task, it may be possible to perform interval bound propagation (Gowal et al., 2018) over multi-modal encoders.

***Robustness to semantics-preserving transformations.*** Ultimately, embeddings should be trained so that semantically close inputs, within and across modalities, are encoded into similar vectors in the embedding space. There is prior work on adversarially robust *perceptual similarity* metrics (Ghazanfari et al., 2023) but perceptual similarity is insufficient for aligned embeddings. It is based on the features of a specific image classifier and only captures semantic similarity insofar as images belong to the same class, i.e., it only aligns images with the labels of that classifier. Alignment in multi-modal embeddings is supposed to cover much broader semantic similarity.

For text, there exist broad classes of semantics-preserving transformations (e.g., substituting words with synonyms), and models can be trained to be robust with respect to word substitutions (Jia et al., 2019; Miyato et al., 2017; Yoo & Qi, 2021). Our attacks exploit adversarial perturbations in images and audio, where semantic similarity is not as well defined as in text. In the image domain, robustness under different distance metrics results in vastly different performance (Tramèr & Boneh, 2019).

To ensure that the embeddings of semantically similar inputs are close to each other, the range of encoders for different modalities should overlap with very fine-grained control over changes in the embedding given a change in the input. With such control, it is not clear how to prevent an adversary from crafting inputs that are closer to the target than organically aligned inputs. Anomaly detection using high-dimensional statistics or enforcing a specific geometry for each modality are interesting topics for future work.

# 6 LIMITATIONS

In this work, we used white-box adversarial perturbation techniques. Therefore, our evaluation is limited to ImageBind, which is the only open-source aligned multi-modal embedding model that is publicly available at the time of this writing. There exist black-box adversarial examples (Guo et al., 2019; Ilyas et al., 2018) but it is not clear how to deploy them in a setting where the adversary only observes outputs of complex downstream models (e.g., diffusion models based on random noise).

For the evaluation, we used currently available downstream generative models. Generative models based on ImageBind embeddings are a new, rapidly evolving field and publicly available implementations are limited. Image generation with BindDiffusion (Lin et al., 2023) uses the unCLIP model (Ramesh et al., 2022) trained on CLIP embeddings (Radford et al., 2021). Therefore, even in the absence of adversarial perturbations, BindDiffusion fails to generate images from the embeddings of many images, sounds, and texts. PandaGPT (Su et al., 2023), although trained on ImageBind embeddings, was fine-tuned only on image-text pairs. Therefore, in some cases it interprets embeddings of sounds as if they were images (see Figure 3). Our attack targets embeddings and is agnostic to downstream models. We expect that improvements in the quality of downstream models will make the attack more effective.

# 7 CONCLUSION

This paper demonstrates that aligned multi-modal embeddings such as ImageBind are highly vulnerable to adversarial perturbations that create *cross-modal illusions*, i.e., inputs in one modality that are aligned with semantically unrelated inputs in another modality. An adversary can fool any downstream task by perturbing a benign input from one modality to match, in the embedding space, an arbitrary, adversary-chosen input from another modality. This attack works across both natural and emergent alignment (i.e., a perturbed sound of a howling wolf appears as an image of a sheep to downstream generative and classification tasks) and reliably produces inputs that appear much closer in the embedding space than semantically similar inputs. Finally, we discuss potential countermeasures, limitations of our current approach, and propose future work.

## ETHICAL CONSIDERATIONS

The only purpose of analyzing vulnerability of multi-modal embeddings to adversarial inputs is to help develop more robust embeddings and motivate research on defenses.

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
