# OpenReview forum: "Ceci n'est pas une pomme: Adversarial Illusions in Multi-Modal Embeddings"
_ICLR.cc/2024/Conference — Submitted to ICLR 2024_

### Official Review · Reviewer_8PvS · 2023-10-28

**Soundness:** 3 good
**Presentation:** 3 good
**Contribution:** 2 fair
**Rating:** 5
**Confidence:** 4

**Summary:**

This paper proposes to use adversarial perturbation to align the perturbed image with a given target text/sound.

**Strengths:**

- This paper studies the vulnerability of multi-modal embedding.
- The experiments also cover the acoustic information.

**Weaknesses:**

- The literature is not well-surveyed, which makes the novelty of the attack not convincing. For example, BadEncoder (S&P'22) already implements such an idea to attack the CLIP model. There are also many following works which cite BadEncoder and study the vulnerability of multi-modal embeddings. How do the authors position the novelty of this work in these literatures?

**Questions:**

Please see the weakness part above.

---

> ### Author Response · Authors · 2023-11-21
>
> BadEncoder is a training-time attack, it assumes that the attacker poisons the training of the model.  In the case of ImageBind or CLIP, it could be used only when the models were trained.  Ours is an evaluation-time attack on the original, unmodified and unpoisoned ImageBind.  It does not involve injecting anything into the training data and does not assume that the attacker has access to the model during training.  We will clarify this fundamental difference between the threat models of backdoor attacks (like BadEncoder) and adversarial inputs (like ours) in Background and Related Work.

---

### Official Review · Reviewer_xAWL · 2023-10-29

**Soundness:** 3 good
**Presentation:** 3 good
**Contribution:** 3 good
**Rating:** 6
**Confidence:** 5

**Summary:**

The paper introduces a new attack called "adversarial illusions" against multi-modal embedding models like ImageBind. These models embed inputs like images, text, and audio into a shared embedding space. The attack involves making small perturbations to an input, like an image, so that its embedding becomes very close to a completely different, adversary-chosen input in another modality, like text. This fools downstream tasks relying on the embeddings, as they now interpret the perturbed input based on the adversary's target instead of the original semantics. Experiments demonstrate the effectiveness of the attack. The authors also discuss potential defenses like adversarial training and certifications. Overall, the work demonstrates serious vulnerabilities in cross-modal alignment of current multi-modal embeddings.

**Strengths:**

This proposes adversarial illusions that are attacks for multimodal embedding models. Given the growing popularity of multi-modal models, this line of research is important and interesting. The idea to adversarially associate a modality with another one unrelated to the semantic of the input is interesting.

**Weaknesses:**

**Main comments:**

While the paper is interesting and this first version is decent, there are a lot of missing experiments that could strengthen the paper and better motivate certain choices:

- The paper proposes to use the I-FGSM attack to create their adversarial illusions:
	- Why use I-FGSM and not PGD, which is the best-known and better attack?
	- The authors could also experiment with DiffPGD, a newer PGD attack based on the diffusion model [1]. The adversarial perturbation is really visible in Figure 7, leveraging diffusion models could improve the attack.
	- Why using the $\ell_\infty$ norm? Have the authors experimented with other norms (e.g. $\ell_2$)?

- The authors seem to have experimented only with cross-modality? Can the authors create adversarial illusions on the same modality?
- Is it possible to investigate the transferability of the attack?  e.g. against other multimodal foundation models?
- The authors seem to be experimenting only with targeted attacks. Would it be possible to maximize the following loss:
$$
\ell = 1 - \cos\left( \theta^{m_1}(x^{m_1}+\delta), \theta^{m_1}(x^{m_1}) \right)
$$


**Other comments:**
- What is the difference between figures 1,3,4,5? It seems only the modality and examples are different. These figures take up a lot of space in the paper, I think these figures could be reduced or some of them could be put in the appendix to leave space for further experiments.
- The authors devote a whole section to countermeasures, but it seems that they do not do any experiments. If the authors focus so much on countermeasures, some experiments should be done.

Xue et al. Diffusion-Based Adversarial Sample Generation for Improved Stealthiness and Controllability

**Questions:**

See weaknesses

---

> ### Author Response · Authors · 2023-11-21
>
> Thank for the review!
>
> 1. I-FGSM comments.
>     1. In general, PGD refers to I-FGSM with a random initialization in an $\ell_\infty$ ball around the source image. This method may perform marginally better for “conventional” adversarial examples, in our setting this benefit is greatly diminished since adversarial alignment must only ‘beat’ organic alignment. We will add an experiment confirming this observation in the **Appendix**.
>     2. As with the above, the $\ell_\infty$ norm (the most common in the literature) is sufficient for an attack that achieves all its objectives, i.e., strong adversarial alignment that fools all downstream tasks. We can add some experiments on other $\ell_p$ norms into the **Appendix**.
>     3. The image in question uses the perturbation bound of $\epsilon = \frac{16}{256}$. As we reported elsewhere in the submission, our attack is effective with much lower bounds. We regenerated the image with a lower bound and it shows no visible noise. While DiffPGD is fascinating, it appears unnecessary in our case since our attack already achieves the desired adversarial alignment with tiny perturbations (DiffPGD can only improve image quality, not performance).
> 2. Same-modality adversarial examples are strictly easier than cross-modal illusions, because, as discussed in Cross-Modal Illusions, the modality gap only arises between modalities. In particular, the unbounded case can be trivially solved within the same modality (i.e., $\delta = \mathbf{a}^{m_2} - x^{m_1}$) whereas a solution may not exist for the cross-modal case. We will provide some experiments in the Appendix.
> 3. To expand the evaluation, we have shown that our attack is effective on AudioCLIP (with a 100% success rate for $\epsilon$ larger than or equal to $\frac{4}{256}$. In addition, we are conducting experiments on the transferability between embedding spaces. The experiments will be added to our **Evaluation** section.
> 4. The targeted attacks we present in the paper are strictly harder than the suggested untargeted attacks. Rather than perturb images to induce *any* non-source class, we align with a *specific* non-source class. The suggested loss function reduces to finding any orthogonal vector in the unbounded case. Since we are in high-dimensional space, sampling randomly is likely to produce an orthogonal vector, and can be computed far more efficiently. In the bounded case, empirically, optimizing our suggested attack with *any* target until the alignment between the perturbed input and the source is less than the organic alignment is sufficient. This can be achieved with extremely small perturbation bounds.
> 5. Thank you for the feedback.   We will move some examples to the Appendix.
> 6. **Add JPEG experiments** Our **Countermeasures** section has been updated to show how our attack evades the JPEG compression defense, a common method to mitigate adversarial perturbations. The JPEG-resistant attack still achieves a $75\\%$ success rate in the presence of the defense. It also evades the defense that checks consistency of embeddings of different input augmentations.  As we note in the paper, there are limitations to most traditional defenses. Developing new defenses specifically for embeddings is an interesting research direction.

---

> > ### Comment · Reviewer_xAWL · 2023-11-22
> >
> > Thank you to the reviewers for their response. Most of my comments have been answered, I'll raise my score from 5 to 6.

---

### Official Review · Reviewer_cUzT · 2023-11-04

**Soundness:** 2 fair
**Presentation:** 3 good
**Contribution:** 3 good
**Rating:** 6
**Confidence:** 3

**Summary:**

This paper demonstrates that the semantic meaning of multi-modal embeddings can be easily manipulated using a simple white-box attack, which is termed adversarial illusion. An attacker only needs to describe in text what they want the input data to mean, and the multi-modal model ImageBind will interpret the attacked but seemingly normal images or audio as conveying the attacker's intended meaning, resulting in cross-modal illusions. This causes ImageBind to make mistakes on downstream tasks even without knowing what these tasks are.

**Strengths:**

- This paper's approach to cross-modal adversarial attacks on images, audio, and text is quite novel.
- The discovery that multi-modal embeddings can be aligned to a target input arbitrarily chosen by an attacker is interesting.
- The paper is mostly well-written and well-organized.

**Weaknesses:**

The main weakness of this paper is the incompleteness of the experiments. There's a lack of experiments involving other models, and the variety of experimental tasks is insufficient. Therefore, I give a 5-point rating initially.
- Using just one multi-modal model, ImageBind, in the experiments to demonstrate that “multi-modal embeddings can be vulnerable to an attack” may be somewhat insufficient. Conducting experiments on AudioCLIP[1], another contrastively pre-trained multi-modal model, would make the claim more convincing.
- Regarding using ImageBind in the experiments, more experiments, e.g. audio classification, image-to-text retrieval, and audio-to-video retrieval, could have been done to strengthen the claim.

[1] Andrey Guzhov, Federico Raue, Jörn Hees, Andreas Dengel. AudioCLIP: Extending CLIP to Image, Text and Audio. ICASSP 2022.

**Questions:**

- I'm uncertain about why black-box attacks cannot be applied in this context. For instance, images and mel spectrograms can still be misclassified into the target input by introducing specific noises calculated by SimBA[2]. Some further clarification regarding the limitations or inapplicability of black-box attacks in this context would be helpful.
- Are the other white-box attacks as effective as I-FGSM in performing adversarial illusions?
- The experiments are conducted using ImageBind, a model that is contrastively pre-trained and projects all modality data into an image embedding space. If ImageBind were replaced with a multi-modal model like ChatBridge[3], which is not contrastively pre-trained and projects all modality data into a text embedding space, would this still demonstrate the vulnerability of multi-modal embeddings to the adversarial illusion attack?
- In Figure 5, could you please clarify why the target input, a sheep image, is encoded by a text encoder rather than an image encoder?
- (Minor) Is the I-FGSM formula complete? I think a clipping operation is missing in it.
- (Minor) I find it a bit confusing whether unCLIP can be used as a generative model. In the "Downstream models" section of Section 2, the paper mentions that "diffusion models that operate on CLIP embeddings, e.g., unCLIP, can also operate on ImageBind embeddings." However, in Section 6, the paper also notes that BindDiffusion, which employs the unCLIP model, struggles to generate images from the multi-modal embeddings. Some clarification on this apparent discrepancy would be appreciated.

[2] Chuan Guo, Jacob R. Gardner, Yurong You, Andrew Gordon Wilson, Kilian Q. Weinberger. Simple Black-box Adversarial Attacks. ICML 2019.
[3] Zijia Zhao, Longteng Guo, Tongtian Yue, Sihan Chen, Shuai Shao, Xinxin Zhu, Zehuan Yuan, Jing Liu. ChatBridge: Bridging Modalities with Large Language Model as a Language Catalyst. arXiv:2305.16103.

---

> ### Author Response · Authors · 2023-11-21
>
> Thank you for the review!
>
> **Weaknesses:**
>
> 1. To expand the evaluation, we have shown that our attack is effective on AudioCLIP (with a 100% success rate for $\epsilon$ larger than 4. In addition, we are conducting experiments on the transferability between embedding spaces. The experiments will be added to our **Evaluation** section.
> 2. We are also conducting experiments on Audio Classification for both ImageBind and AudioCLIP. We expect the numbers to be good, considering audio retrieval is a strictly harder desiderata than classification. The experiments will be added to our **Evaluation** section.
>
> **Questions:**
>
> 1. Black-box attacks are interesting future work but there are some inherent challenges to this formulation that we will add to the **Limitations** section. Consider two alternate definitions of ‘black-box’: (1) the attacker has no access to the internal representations of the encoder, but does have access to the resulting embeddings, and (2) the attacker has only access to the outputs of a specific downstream task. Recall that our attack is targeted.
>
>     For (1), satisfying the classification constraint (i.e., changing the label to a specific target) is a significantly easier task than finding an image that is arbitrarily close, in a high-dimensional embedding space, to a specific target embedding. Modern black-box techniques that rely on Bayesian Optimization are particularly ill-suited for this high-dimensional objective. That said, this is an interesting topic for future work.
>
>     For (2), while it is possible that an adversarial example generated for a specific downstream task may generalize to others, the representation-merging of the example and its target could happen in the downstream model as opposed to the encoder, making generalization to all downstream tasks hard to achieve. This is also a promising direction for future work.
>
> 2. We demonstrate that in our setting, I-FGSM’s performance is essentially perfect, achieving high adversarial alignment (much higher than organic alignment) with tiny perturbations.  We are not sure about the motivation for exploring other methods but can evaluate PGD if requested.
> 3. Given the strength of our adversarial alignment, we expect that any system that relies on a unified embedding space, regardless of whether it was contrastively trained, will be vulnerable to attack. In particular, our attack relies on the former as opposed to the latter.
> 4. Thanks for pointing that out! That should read ‘Image Encoder’.
> 5. Thanks again, we are missing the clipping operation.
> 6. We will clarify this distinction in the **Limitations** section. To rephrase, ImageBind embeddings work on unCLIP, but seemingly perform worse. We hope to disentangle our observations on the alignment of our attack and the limitations (due to training) of the models we use.

---

> > ### Comment · Reviewer_cUzT · 2023-11-23
> >
> > Thank you to the authors for addressing my questions and resolving most of my concerns. If the authors can present the experimental results of AudioCLIP and audio classification here or in the paper, and if these results also show the effectiveness of adversarial illusion, I would raise my score to 6.

---

> > > ### Author Response · Authors · 2023-11-23
> > > **AudioCLIP results**
> > >
> > > Thank you for the feedback, please see the attached table (based of Table 1 measuring zero-shot accuracy) with comparison to AudioCLIP. Overall, we have observed a similar behavior and the strength of our attack.
> > >
> > > | Alignment     | $\epsilon_V$ | Top-1 (ImageBind) | $\mathrm{align}(\cdot)$ (ImageBind) | Top-1 (AudioCLIP) | $\mathrm{align}(\cdot)$ (AudioCLIP) |
> > > |---------------|--------------|-------------------|------------------------------------|-------------------|-------------------------------------|
> > > | Organic       | -            | 67%               | 0.2878 ± 0.005                     | 27%               | 0.1445 ± 0.003                      |
> > > | Adversarial   | 0            | 0%                | 0.0892 ± 0.005                     | 0%                | 0.0670 ± 0.003                      |
> > > |               | 1            | 93%               | 0.5741 ± 0.015                     | 90%               | 0.3560 ± 0.008                      |
> > > |               | 4            | 100%              | 0.8684 ± 0.006                     | 100%              | 0.6510 ± 0.009                      |
> > > |               | 8            | 100%              | 0.9241 ± 0.004                     | 100%              | 0.7323 ± 0.007                      |
> > > |               | 16           | 100%              | 0.9554 ± 0.002                     | 100%              | 0.7841 ± 0.006                      |
> > > |               | 32           | 100%              | 0.9692 ± 0.001                     | 100%              | 0.8074 ± 0.006                      |
> > >
> > > We will integrate these results into the paper.

---

> > > > ### Comment · Reviewer_cUzT · 2023-11-23
> > > >
> > > > Thank you for providing the AudioCLIP experiment. How about the audio classification experiment?

---

> ### Author Response · Authors · 2023-11-23
>
> We are still in the process of getting those numbers, but in small scale experiments on ImageBind for $\epsilon = 0.05$ we achieve 100% accuracy. We will report the expanded results for these experiments in the final version of the paper with the remaining choices of $\epsilon$ as well as equivalents on AudioCLIP. Given this result and the efficacy on audio retrieval, we project that the results will be better than or equal to the audio retrieval experiments. Recall that zero-shot retrieval requires the identification of an arbitrary caption related to an audio while classification seeks out labels that the model was already trained on (in the case of ImageBind). Importantly, in the space of multimodal encoders, the mechanics of the two tasks are equivalent: finding the label or caption nearest in embedding space to the input. Since, our attack works on arbitrary (audio, caption) pairs, we believe that the attack will be effective on (audio, label) pairs as well.
>
> (edits: 1. a previous version of this comment mentioned the wrong epsilon bound and 2. added details about retrieval.)

---

> > ### Comment · Reviewer_cUzT · 2023-11-23
> >
> > Thank you for informing the current progress. Since my concerns have been addressed, I will raise the score to 6 as I promised.

---

### Official Review · Reviewer_Tpfm · 2023-11-10

**Soundness:** 2 fair
**Presentation:** 3 good
**Contribution:** 2 fair
**Rating:** 3
**Confidence:** 5

**Summary:**

This paper studies adversarial attacks for images and audio using multimodal embeddings. The method builds upon a pretrained multimodal embedding model such as ImageBind, and can be used to attack downstream models that also use this model as the embedding model. Given an image/audio and an adversarial text, the adversarial attack is applied to the image/audio space to maximize the cosine similarity between the image/audio embedding and the adversarial text embedding. Experiments show that adversarial examples can fool downstream tasks that use the same embedding model.

**Strengths:**

1. Experiments include image attacks and audio attacks, which are more comprehensive than previous works that mostly experiment with one modality.
2. The paper is well-written.

**Weaknesses:**

1. The downstream task uses exactly the same embedding model as the one being attacked. Therefore, it is not surprising that they can be fooled. It would be more interesting if some unexpected findings/insights were provided.
2. As the authors acknowledged, several existing papers have studied adversarial attacks for multimodal learning (e.g., for CLIP and for multimodal large language models). More thorough comparisons with the previous works should be done.

**Questions:**

How robust is the attack to adversarial defense?

---

> ### Author Response · Authors · 2023-11-21
>
> Thank you for your review.
>
> **Weaknesses:**
>
> 1. Ours is the first attack that targets multi-modal embeddings, as opposed to a specific task, and is completely agnostic of downstream tasks.  Success of an embedding attack cannot be evaluated without measuring its effect on tasks that use the embedding.
>
>     To expand the evaluation, we have shown that our attack is effective on AudioCLIP (with a $100\\%$ success rate for $\epsilon$ larger than 4. In addition, we are conducting experiments on the transferability between embedding spaces. The experiments will be added to our **Evaluation** section.
>
>     Previous attacks on multimodal learning target a single modality and don’t need to deal with modality gap. We emphasize our main observation: *adversarial* alignment can be made significantly closer than any *organic* alignment regardless of input and target modality. As a result, our attack affects **any current and future downstream task** based on the attacked embedding (or, as we show in our new experiments, similar embeddings). With an increased industry focus on large, centralized encoders, the same embedding will underpin many downstream use cases.  We show they can be attacked wholesale, without the attacker even knowing what they are.
>
> 2. Previous work targeted specific tasks, not multi-modal embeddings themselves. Our updated **Background and Related Work** section explains how our attack relates to [1, 2, 3], which were helpfully provided by our other reviewers.
>
> **Questions:**
>
> 1. Our **Countermeasures** section has been updated to show how our attack evades the JPEG compression defense, a common method to mitigate adversarial perturbations. The JPEG-resistant attack still achieves a $75\\%$ success rate in the presence of the defense. It also evades the defense that checks consistency of embeddings of different input augmentations.

---

> > ### Comment · Reviewer_Tpfm · 2023-11-23
> > **Similar idea explored by ICLR 2023 paper "Understanding Zero-Shot Adversarial Robustness for Large-Scale Models"**
> >
> > I respectfully disagree with the claims that (1) it is the first attack that targets multi-modal embeddings and (2) previous work targeted specific tasks but not multi-modal embeddings themselves. Many previous works have attacked the CLIP model. For instance, why is the current manuscript fundamentally different from this work from ICLR 2023 [1], except that the audio experiments are included?
> >
> > [1] Chengzhi Mao, Scott Geng, Junfeng Yang, Xin Wang, Carl Vondrick, **Understanding Zero-Shot Adversarial Robustness for Large-Scale Models** (ICLR 2023)

---

> ### Author Response · Authors · 2023-11-23
> **re ICLR'23 paper**
>
> Thank you for providing the reference, we want to point that the attack defined in **Sec 3.1** of the paper is an attack on a *specific* downstream task (zero-shot image classification), not a **downstream-task agnostic** attack on the multi-modal embedding that affects all tasks based on that embedding (which is our contribution in this submission) including **generative** tasks that don’t assign a fixed label to the input.
>
> We, therefore, argue that our attack is the first downstream-task agnostic attack unlike the method presented in that paper.
>
> | Input type                 | Top-1         |          | Top-5         |          |
> |----------------------------|---------------|----------|---------------|----------|
> |                            | Original label| Target label | Original label| Target label |
> | Original image x           | 85%           | 0%       | 99%           | 0%       |
> | Adversarial illusion xa    | 77%           | 0%       | 95%           | 0%       |
> | Generated(ImageBind(x))    | 42%           | 0%       | 64%           | 2%       |
> | Generated(ImageBind(xa))   | 0%            | 64%      | 1%            | 92%      |

---

### Meta-Review · Area_Chair_sNjG · 2023-12-09

**Metareview:**

**Summary**

This paper propose "adversarial illusions" to attack the multi-modal embeddings. It is shown that an attacked image can be encoded to an embedding that's similar to any target text, and the performance of downstream tasks is severely decreased.

**Strengths**

- Adversarial attack to a multi-modal model covering image, audio, and text is new
- Multiple reviewers like the presentation of this paper. However, I'd suggest the authors to put more informative captions for the figures, for easier and faster understanding the high-level ideas.

**Weaknesses**

- Lack of quantitative experiments. All reviewers agreed (in a private discussion) that this paper lacks quantitative experiments, compared to average ICLR / similar conference papers.
- Technical contribution is limited. The community already know adversarial attack is easy for modern neural networks, especially for white box attack. Researchers do not expect multi-modal foundation models could bring any change to this belief, and the conclusion of this paper did not, either.

**AC's additional comments**

I understand the authors' argument that attacking the shared embedding space for three modalities is new, and text-to-image attack has been under-explored. However, I agree with reviewer Tpfm that the technical contribution of this paper is low. Multi-modal foundation models are not much different from models ~5 years ago in terms of adversarial robustness, and I don't think the community expect they are. Besides, I'm not surprised that image generation can also be attacked, because it is a more difficult task than recognition. Therefore, the findings in the submission seems not significant enough for ICLR in my opinion.

Reviewer cUzT raised the ethical flag and noted: "this paper finds that multi-modal models are vulnerable to a simple adversarial attack and doesn't show how this attack can be defended against, malicious people might exploit this vulnerability and attack the applications of these multi-modal models". I think this is the nature of adversarial attack papers, and the authors have discussed the ethical concerns, thus no ethics review required.

The uninformative review by 8PvS, who is also non-responsive, is completely ignored in my decision process.

**Justification For Why Not Higher Score:**

I acknowledge that some findings of this submission is new, and no correctness issues are found. However, this submission does not bring significant enough technical contribution / insights for the community (see the meta review). Hence, I recommend reject.

My suggestion for the authors is to perhaps work on adversarial defense techniques for multi-modal models. In adversarial learning, defense has been much more challenging than attack historically, but on the other hand more rewarding. Perhaps the community would appreciate more if multi-modal models can bring new possibilities in adversarial defense.

**Justification For Why Not Lower Score:**

N/A

---

### Decision · Program_Chairs · 2024-01-16

Reject